# Association of Social Frailty with Physical Health, Cognitive Function, Psychological Health, and Life Satisfaction in Community-Dwelling Older Koreans

**DOI:** 10.3390/ijerph18020818

**Published:** 2021-01-19

**Authors:** Hana Ko, SuJung Jung

**Affiliations:** 1College of Nursing, Gachon University, Incheon 21936, Korea; hanago11@gachon.ac.kr; 2College of Nursing, Seoul National University, Seoul 03080, Korea

**Keywords:** social frailty, older adults, life satisfaction

## Abstract

Social frailty affects various aspects of health in community-dwelling older adults. This study aimed to identify the prevalence of social frailty and the significance of its association with South Korean older adults’ health status and life satisfaction. This study involved a secondary data analysis of the 2017 National Survey of Older Koreans. From the 10,299 respondents of the survey, 10,081 were selected with no exclusion criteria. Multiple regression analyses were conducted to identify the factors related to life satisfaction. Compared with the robust and social prefrailty groups, the social frailty group had higher nutritional risk (χ² = 312.161, *p* = 0.000), depressive symptoms (χ² = 977.587, *p* = 0.000), cognitive dysfunction (χ² = 25.051, *p* = 0.000), and lower life satisfaction (F = 1050.272, *p* = 0.000). The results of multiple linear regression, adjusted for sociodemographic and health-related characteristics, indicated that social frailty had the strongest negative association with life satisfaction (*β* = −0.267, *p* = 0.000). However, cognitive function was significantly positively associated with life satisfaction (*β* = 0.062, *p* = 0.000). Social frailty was significantly correlated with physical, psychological, and mental health as well as life satisfaction in community-dwelling older South Koreans. Therefore, accounting for the social aspect of functioning is an essential part of a multidimensional approach to improving health and life satisfaction in communities.

## 1. Introduction

The median age of the South Korean population is rapidly increasing; in 2019, individuals aged 65 and over accounted for 14.9% of the South Korean population, and this proportion has been projected to exceed 46.5% by 2067 [1]. Given this rapid increase in the older adult population, frailty and life satisfaction in this age group are becoming more critical than at any other age [2,3]. A high level of life satisfaction is an indicator of happiness and success in old age [4,5,6]. Life satisfaction, a subjective evaluation of contentment with one’s life [4], is affected not only by intrinsic physical and mental capacities but also by functional abilities and environmental aspects such as social factors [7]. Therefore, the World Health Organization’s initiative to create age-friendly cities includes measures to increase older adults’ life satisfaction [8]. Creating age-friendly environments requires collaboration and coordination across multiple sectors and with diverse stakeholders, including older people. The foundation of such efforts is to allow older adults to have social relationships in their own life community.

Frailty is defined as a biological syndrome of extreme vulnerability to endogenous and exogenous stressors associated with multisystem decline in physiological reserve, resulting in increased risk of adverse outcomes including disability, hospitalization, and death [3,9]. This multidimensional concept has physical, cognitive, psychological, and social components [10].

Recently, the concept of “social frailty” has been increasingly emphasized. Based on a scoping review [4] and using the theory of social production function, a conceptual framework has been proposed. Social frailty is defined as a continuum of being at risk of losing, or having lost, resources that are important for fulfilling one or more basic social needs during the lifespan [11]. However, existing explorations of social frailty have been complicated by the interconnections of contextual, societal, and cultural considerations [10].

Previous studies have reported that social frailty is associated with muscle weakness [12] and cognitive function [2], and that social frailty can lead to disability and mortality [13]. In a study of homeless women, drug use, emotion regulation, and daily alcohol use were significant correlates of social frailty [14]. However, there have been few reports of the prevalence of social frailty and how it relates to older South Korean adults’ health status and life satisfaction.

Thus, this study aims (1) to examine the differences in general and health status characteristics of community-dwelling older adults in South Korea according to social frailty status; (2) to examine the correlations among social frailty status, nutritional status, depression, cognitive function, and life satisfaction; and (3) to explore the health-related predictors associated with life satisfaction.

## 2. Materials and Methods

### 2.1. Participants

This cross-sectional study used secondary data from the 2017 National Survey of Older Koreans [15]. The National Survey of Older Koreans, conducted by the Ministry of Health and Welfare every three years, seeks to gather the data necessary to devise policy measures to improve quality of life and better manage population aging in this age group. The 2017 survey included 10,299 individuals aged 65 or older living in standard residential facilities or premises in 17 metropolitan cities and provinces across South Korea. The 2017 National Survey of Older Koreans sample was selected using a proportional two-stage stratified sampling method, which was first stratified and collected by 17 metropolitan cities and provinces across Korea and then again by neighborhoods in the nine provinces and Sejong (but not in the metropolitan cities) [15]. The Ministry of Health and Welfare research team applied various weights in the raw data to ensure the accuracy of estimations. The weight of the raw data was adjusted by considering the weights for households and individuals [15]. The data were obtained through in-person interviews in 934 survey areas from 12 June to 28 August 2017. The survey was conducted by 60 surveyors, trained by the research staff in advance. Surveyors checked the answered questionnaires for any omissions and errors and relayed their feedback to the research team. Raw data used in this study were obtained on 5 June 2020 after obtaining approval from the Health and Welfare Data Portal (https://data.kihasa.re.kr/). From the 10,299 respondents of the 2017 National Survey of Older Koreans, 10,081 were selected without any exclusion criteria; 218 were excluded for missing responses.

### 2.2. Measures

#### 2.2.1. Sociodemographic and Health-Related Characteristics

Sociodemographic characteristics included age, gender, education level, economic status, and living conditions (living alone, living with partner, living with others). Economic status was sorted in ascending order by annual personal income and divided into five categories so that each group contained 20% of the participants. Then, only the bottom 20% of the group were used in the analysis, as they best suited our interests. Health-related characteristics included the number of prescription medications, diagnosis of chronic diseases, subjective health status (very healthy, healthy, average, in ill health, in very ill health), lower-extremity muscle (sitting and standing up), and lifestyle (smoking and physical activity).

#### 2.2.2. Social Frailty

To identify and assess social frailty, we operationalized the concept into five categories based on a previous study [9]: going out (not participating in any leisure and social activities such as travel, hobbies, learning or studying, social clubs, networking, political and social groups, volunteering, senior citizen centers, community centers for older adults), visiting friends (no), feeling worthless (yes), living alone (yes), and contact with someone (no). Participants with none, one, and two or more of these components were classified into the robust, social prefrailty, and social frailty groups, respectively [9,16,17].

#### 2.2.3. Nutritional Status

Nutritional status was measured using “Determine Your Nutritional Health,” a tool developed by the Nutrition Screening Initiative [18]. Used to assess nutritional status in older adults, this instrument consists of 10 items, each of which is rated from 1 to 4. The range of possible scores is from 0 to 21; accordingly, nutritional status is categorized as good (0 to 2 points), moderate risk (3 to 5 points), and high risk (6 points or more). In this study, 3 points or more were classified into the nutritional risk group.

#### 2.2.4. Depression

Depression was measured using the 15-item Geriatric Depression Scale-Short Form Korean Version (GDSSF-K) [19]. The GDSSF-K includes five positive items and 10 negative items in a yes/no response format. The total GDSSF-K score was obtained by counting the number of “yes” responses after the positive items were reverse coded so that higher scores indicated higher levels of depressive symptoms. The total scores ranged from 0 to 15. The criteria for determining depression were “normal” in the total score for less than 5 points, “moderate depression” for 6 to 9 points, and “depressed” for more than 10 points. In the 2017 National Survey of Older Koreans [15], people scoring more than 8 points were classified as “depressed”, and this criterion was used in our analysis. There is evidence supporting the construct and criterion related validity of the GDSSF-K [19]; Cronbach’s alpha was 0.88 in a previous study [19] and 0.89 in the present study.

#### 2.2.5. Cognitive Function

Cognitive function was measured using the Mini-Mental State Examination for Dementia Screening (MMSE-DS) [20]. The 19-item MMSE-DS has a maximum score of 30 points, with higher scores indicative of higher cognitive function. This tool has been standardized by age, gender, and educational level for normative cognitive function assessment in older adults in South Korea [20]. There is evidence supporting the validity of the MMSE-DS [21]; Cronbach’s alpha was 0.82 in a previous study and 0.93 in the present study.

#### 2.2.6. Life Satisfaction

Life satisfaction was measured using the question “To what extent are you satisfied with the following aspects of your life: health status, economic status, relationship with spouse, relationship with children, leisure and cultural activities, and relationships with friends and society?” The response options were: 1 = very satisfied, 2 = satisfied, 3 = average, 4 = not satisfied, and 5 = not satisfied at all. Responses to all items were reverse coded, so that higher scores indicated higher levels of life satisfaction. The total scores ranged from 6 to 30; Cronbach’s alpha was 0.61 in the present study.

### 2.3. Ethical Considerations

The 2017 National Survey of Older Koreans was approved by Statistics Korea (Approval No. 11771). For our study, after obtaining approval from the Korea Institute for Health and Social Affairs, we received raw data without personal identification information. Moreover, the study was approved by the Institutional Review Board (IRB No.: 1044396-202006-HR-110-01) of Gachon University, to which one of the researchers is affiliated.

### 2.4. Data Analyses

Sample characteristics were summarized using means and standard deviations (SDs) for continuous variables and proportions for categorical variables. To evaluate the differences in characteristics between participants from the three groups (robust, social prefrailty, and social frailty), we used Pearson’s χ^2^ test for categorical data and analysis of variance for continuous data. Then, we performed multiple regression analysis to identify the factors related to life satisfaction. Before running the multiple regression analysis, we conducted a correlation analysis, and the independent variables were tested for multicollinearity using tolerance value and variance inflation factor (VIF). Less than 10% of confirmed missing cases were excluded from the analysis by applying listwise deletion [22]. All statistical analyses were conducted using SPSS version 23.0 (IBM Corp., Armonk, NY, USA) with the two-tailed significance level set at 0.05. For effect size, we followed Cohen’s criteria (0.10 = small, 0.25 = medium, 0.40 = large) [23] in analysis of variance, and Rea and Parker’s criteria (0.00 ≤ x < 0.10 = negligible, 0.10 ≤ x < 0.20 = weak, 0.20 ≤ x < 0.40 = moderate, 0.40 ≤ x < 0.60 = relatively strong, 0.60 ≤ x < 0.80 = strong, 0.80 ≤ x ≤ 1.00 = very strong) [24] in Pearson’s χ^2^ test.

## 3. Results

### 3.1. General Characteristics of the Study Sample

The sample characteristics are shown in Table 1. Out of the 10,081 participants, 1292 (12.8%), 4281 (42.5%), and 4508 (44.7%) were classified into the robust, social prefrailty, and social frailty groups, respectively. The mean age in the robust, social prefrailty, and social frailty groups was 72.2 years, 73.9 years, and 75.6 years, respectively. Those in the social frailty group were older, had lower education level, economic status, and subjective health status, and were more likely to live alone compared with the robust and social prefrailty groups (*p* = 0.000). Regarding lifestyle, there was a difference in physical activity according to social frailty status (*p* = 0.000). However, there were no statistically significant differences between the three groups regarding smoking.

Additionally, the social frailty group had a higher proportion of participants with more than three diagnosed chronic diseases and a higher number of prescribed medications compared to those in the robust and social prefrailty groups.

### 3.2. Differences in Nutritional Status, Depression, Cognitive Function, and Life Satisfaction by Social Frailty Status

Table 2 displays the differences in nutritional status, depression, cognitive function, and life satisfaction by social frailty status, categorized into the three groups robust, social prefrailty, and social frailty. The prevalence of nutritional risk, depression, and lower cognitive function was highest in the social frailty group. Moreover, the social frailty group scored the lowest on all six categories of life satisfaction: health status, economic status, relationship with spouse, relationship with children, leisure and cultural activities, and relationships with friends and society.

### 3.3. Correlation of Predictors

Table 3 shows that the predictors were correlated. Depression and nutritional status had a high correlation (r = 0.455, *p* = 0.000). In addition, life satisfaction, which was the criterion variable, was significantly correlated with all predictors.

### 3.4. Multiple Regression Analysis

Multiple linear regression analyses revealed that social frailty had the strongest negative association with life satisfaction (*β* = −0.267, *p* = 0.000) (Table 4). However, cognitive function was significantly positively associated with life satisfaction (*β* = 0.062, *p* = 0.000). The variance inflation factor (VIF) of predictors and the tolerance of predictors were 1.427–2.749 and 0.364–0.701 respectively, which suggests the absence of multicollinearity between the predictors.

## 4. Discussion

The prevalence of social frailty in the sample was 44.7%. Further, the social frailty group displayed the highest prevalence of nutritional risk, depression, low cognitive function, and poor life satisfaction. Additionally, social frailty displayed the strongest negative association with life satisfaction. These findings suggest that social frailty may affect older adults’ physical, cognitive, and psychological functions as well as life satisfaction.

The prevalence of social frailty in this study was higher than the rates reported by Tsutsumimoto et al. (11.1%) [9] and Yamada and Arai (18%) [13]. Our social frailty index was operationalized based on several previous studies [9,16,17]. Additionally, our study samples were composed of community-dwelling older adults without any exclusion criteria like disabilities in activities of daily living or severe diseases [9] and long-term care recipients [13]. Therefore, the major differences from previous studies are with regard to the items of the social frailty questionnaire and the study population. While the five-item social frailty assessment is popular [9,25,26,27], participants’ culture and environment have not always been considered in existing research. Regarding other measures of social frailty, as in this study, many previous studies have used participants’ living status (living alone or with someone) [9,12,13,16,17]. In Korea, the proportion of single-person households ages 65 or older is expected to be 36.6% in 2045, the highest compared to couple households (30.2%) or couples and children (9.2%) [28]. Therefore, the country needs to prepare for social frailty caused by population aging and the rapid increase in the number of older adults living alone.

This study shows that older adults with social frailty tend to be more vulnerable to impairments in cognitive and psychological functions than their robust counterparts. This result is in line with those of several previous studies, which revealed that higher levels of social frailty were associated with cognitive dysfunction [9,25] and depressive symptoms [16,25]. Furthermore, deteriorations in social frailty status have been associated with worsening physical nutritional status. Malnutrition is considered one of the physical functions responsible for sarcopenia, osteoporosis, and impaired immune response [29]. In terms of physical functions, recent cohort studies have focused on the impact of social frailty on physical frailty [12] and disability [26,30]. Similarly, social frailty has been shown to be associated with cognitive impairment, depression, and physical functioning in China [25]. These findings confirm that older adults with poor social relationships and social engagement are at an increased risk of multidimensional dysfunctions. Future studies should further delineate the causal relationship between social frailty and multidimensional health functions.

To the best of our knowledge, this is the first study to examine the association between social frailty and life satisfaction in older adults. According to the results, social frailty had a stronger negative association with life satisfaction than with physical, psychological, and cognitive functions. Previous studies have revealed a negative relationship between depression levels and life satisfaction [31,32,33]. Further, in a longitudinal study, long-term life dissatisfaction predicted the onset of major depressive disorder [34]. In addition, cognition has been shown to be associated with life satisfaction [33]. Our study provides a starting point for examining the impact of social frailty on life satisfaction in older adults. Life satisfaction, a subjective cognitive evaluation of an individual’s life [5], is a component and crucial indicator of quality of life [6]. The World Health Organization also emphasizes the importance of social relationships for older adults; the maintenance of social relationships is a prerequisite for healthy aging, bringing life satisfaction [8]. Especially, during the current coronavirus disease 2019 pandemic, social isolation may amplify the prevalence of social frailty in older adults [35]. Therefore, the result that social frailty had the strongest association with life satisfaction in older adults provides justification for preparing plans to increase life satisfaction by preventing social frailty and promoting social relationships.

The strengths of this study are the large sample size and the use of an operationalized assessment to identify social frailty. On the other hand, the large sample size can cause overpower problems in situations where there is little association between groups. It is necessary to pay attention to the interpretation of the results based on the effect size [23,24], which are reported in Table 1 and Table 2. In the difference in characteristics according to social frailty status, the effect sizes of gender, chronic disease, age, number of prescribed medicines, physical activity, subjective health status, nutritional status, and cognitive function were lower than those of other characteristics [24]; however, there was a considerable difference in life satisfaction according to social frailty [23]. While the effect size can confirm the practical difference (or association) between groups [36], it has a limitation in that it is a relative value that can vary depending on the characteristics of the population or other variables [23]. Nevertheless, to our knowledge, the current study is the first to report an association between life satisfaction and social frailty.

However, this study also has some limitations. First, owing to the use of cross-sectional secondary data, causality could not be explored; this should be clarified in a future prospective study. Second, although Diener et al.’s Satisfaction with Life Scale [37] is the most widely used instrument in the field, we used the six categories of life satisfaction surveyed in the 2017 National Survey of Older Koreans. Third, the categories of social frailty were based on recent studies [9,16,17], not an established method. Therefore, future research on tool development to measure social frailty is needed. Lastly, social frailty can be affected by many environmental factors based on social context. Therefore, research is needed to identify the relationship between social frailty and life satisfaction in people of various ethnic groups.

## 5. Conclusions

Social frailty and its association with nutritional status, depression, cognitive function, and life satisfaction should be considered as an integrative comprehensive older adult care. Further studies are needed to develop of efficient social frailty intervention strategies to improve and enhance life satisfaction.

## Figures and Tables

**Table 1 ijerph-18-00818-t001:** General characteristics of participants in the robust, social prefrailty, and social frailty groups.

Characteristic	Total *n* = 10,081	Robust *n* = 1292	Social Prefrailty *n* = 4281	Social Frailty *n* = 4508	χ²/F (*df*)	*p* (E.S.)
Age, years Mean (SD)	74.5 (6.2)	72.2 (5.5)	73.9 (6.0)	75.6 (6.4)	180.424 ^a^ (2)	0.000 (0.139) ^b^
Men *n* (%)	4046 (40.1)	598 (46.3)	1864 (43.5)	1584 (35.1)	87.868 (2)	0.000 (0.093) ^c^
Education, years Mean (SD)	6.8 (4.6)	7.6 (4.3)	7.2 (4.5)	6.1 (4.7)	106.257 ^a^ (2)	0.000 (0.212) ^b^
Low economic status *n* (%)	2037 (20.2)	53 (4.1)	488 (11.4)	1496 (33.2)	1063.055 (8)	0.000 (0.230) ^c^
Living alone *n* (%)	2552 (25.3)	0 (0.0)	450 (10.5)	2102 (46.7)	2061.501 (4)	0.000 (0.320) ^c^
More than 3 diagnosed chronic diseases *n* (%)	5326 (52.8)	556 (43.0)	2135 (49.9)	2635 (58.5)	137.111 (6)	0.000 (0.082) ^c^
Number of prescribed medicines Mean (SD)	4.0 (3.4)	3.3 (3.2)	3.7 (3.2)	4.5 (3.5)	88.864 ^a^ (2)	0.000 (0.132) ^b^
Current smoker *n* (%)	950 (9.4)	129 (10.0)	368 (8.6)	453 (10.0)	5.975 (2)	0.050 (0.024) ^c^
Weakness in lower-extremity muscles *n* (%)	2148 (21.3)	107 (8.3)	682 (15.9)	1359 (30.1)	414.594 (2)	0.000 (0.203) ^c^
No physical activity *n* (%)	3346 (33.2)	289 (22.4)	1263 (29.5)	1794 (39.8)	183.198 (2)	0.000 (0.135) ^c^
Subjective health status, very poor health *n* (%)	469 (4.7)	30 (2.3)	112 (2.6)	327 (7.3)	439.035 (8)	0.000 (0.148) ^c^

Note: E.S. = effect size; ^a^ analysis of variance; ^b^ Effect size *f*; ^c^ Cramer’s *V*.

**Table 2 ijerph-18-00818-t002:** Prevalence of social frailty in different health domains.

Health Domain	Total *n* = 10,081	Robust *n* = 1292	Social Prefrailty *n* = 4281	Social Frailty *n* = 4508	χ²/F (*df*)	*p* (E.S.)
Nutritional status risk ^a^ *n* (%)	6213 (61.6)	623 (48.2)	2401 (56.1)	3189 (70.7)	312.161 (2)	0.000 (0.176) ^e^
Depressed ^b^ *n* (%)	2177 (21.6)	73 (5.7)	495 (11.6)	1609 (35.7)	977.587 (2)	0.000 (0.311) ^e^
Cognitive dysfunction ^c^ *n* (%)	260 (2.6)	10 (0.8)	104 (2.4)	146 (3.2)	25.051 (2)	0.000 (0.050) ^e^
Life satisfaction Mean (SD)	18.81 (3.81)	21.14 (3.09)	19.89 (3.32)	17.06 (3.68)	1050.272 ^d^ (2)	0.000 (0.462) ^f^

Note: ^a^ Determine Your Nutritional Health tool developed by the Nutrition Screening Initiative; ^b^ Geriatric Depression Scale-Short Form Korean Version (GDSSF-K); ^c^ Mini-Mental State Examination for Dementia Screening (MMSE-DS); ^d^ analysis of variance; ^e^ Cramer’s *V*; ^f^ effect size *f*; *df* = degrees of freedom; E.S. = effect size.

**Table 3 ijerph-18-00818-t003:** Correlations between predictors and life satisfaction.

	Social Frailty	Nutritional Status	Depression	Cognitive Function
	*r (p)*	*r (p)*	*r (p)*	*r (p)*
Social frailty				
Nutritional status	0.236 (0.000)			
Depression	0.339 (0.000)	0.455 (0.000)		
Cognitive function	−0.175 (0.000)	−0.261 (0.000)	−0.273 (0.000)	
Life satisfaction	−0.408 (0.000)	−0.486 (0.000)	−0.545 (0.000)	0.339 (0.000)

**Table 4 ijerph-18-00818-t004:** Factors related to the life satisfaction of older adults.

Predictors	Criterion: Life Satisfaction R^2^ = 0.49, *p* = 0.000				
	*β*	SE	*T*	*p*	VIF
Social frailty	−0.267	0.091	−22.422	0.000	2.749
Depression	−0.224	0.008	−24.632	0.000	1.597
Nutritional status	−0.186	0.010	−21.718	0.000	1.427
Cognitive function	0.062	0.009	7.041	0.000	1.528

Note: *n* = 9833 (missing cases are excluded listwise); *β* = standardized coefficient; SE = standard error; VIF = variance inflation factor.

## Data Availability

The authors have no authority over the data, and the data is provided upon request to the Ministry of Health and Welfare.

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
