# Peer review of "Association of Social Frailty with Physical Health, Cognitive Function, Psychological Health, and Life Satisfaction in Community-Dwelling Older Koreans"

_ijerph, 2021, doi:10.3390/ijerph18020818_

Round 1

Reviewer 1 Report

A Review of Social Frailty…….

The article is based on a large-scale study and is generally well-executed. There are some suggestions for revising the article.

  1. Aims of the study, p, 2 lines 54 plus: The aims are stated very broadly and do not reflect the nature of analyses conducted. For example, the authors state: to 1) understand differences in the associations of social frailty with physical, cognitive, and psychological health status and life satisfaction and 2) to compare the factors related to social frailty and life satisfaction.

The first aim suggests that the authors would examine correlations of social frailty with physical, cognitive,… and life satisfaction and test if these correlations differ significantly.  This was never done in the study. No correlations or differences in the associations were presented, what was done was to predict life satisfaction using social frailty, depression, nutritional status & cognitive function (Table 3). Mean Differences were examined comparing frailty groups on depression, cognitive dysfunction, and life satisfaction (Table 2), and physical activity, subjective health, weakness of lower muscle extremity (Table 10). Not much is given in the Sociodemographic section as to how the variables physical, subjective, and muscle weakness were measured. Also, the second aim is vague and it is not clear what the authors mean “factors related to” mean.  There is no mention of nutritional status.

  1. Materials and Methods.

Subjects—I am not clear on the journal’s policy, but the current word is Participants and not Subjects.

  1. p. 3. Line 97. “The GDSSF-K has established construct validity; Cronbach’s alpha was .88 in a previous study [19] and .89 in the present study.” The word “established” is too strong—perhaps better to say there is evidence supporting the construct validity of ….(and provide reference) (the phrase in repeated in other places).
  2. p. 3. Line 112. Provide reliability coefficient for Life satisfaction measure—since you are deriving an overall score.
  3. p. 3. Line 125, Since the 2017 National Survey of Older Koreans used a stratified two-stage cluster sampling method, the weight values considered at sample (something is missing in the sentence: what does weight values mean and how did it benefit you sampling?)

6.p. 4., line 141:  Regarding lifestyle, the social frailty group was did not physical activity than the robust and social prefrailty groups. (grammar?)

  1. Table 1; why say ∓ SD? What does this mean. You are not computing Confidence intervals. Why not just put SD in parenthesis?

Also, given the sample size is very large, effect sizes need to be reported for all chi-squared and ANOVA results in Tables 1 and 2 and used in interpreting the results. It seems some p values are missing (e.g., Normal has a chi-squared value, but no p-value). Also, why not report the exact p-value instead of p < .05? for Nutritional Risk, the chi-squared value is missing. Or, is the Chi-square test for a 3X3 table? Provide df for both chi-squared and F tests.

  1. Multiple-Regression analysis. Social Frailty has a beta coefficient of -.267, but depression has beta = .224; these are not that different. Comparing beta weight based on the absolute value is problematic when the variables are intercorrelated. Given that R squared = beta1(ryx1) + beta 2(ryx2) for a two-variable case, we need to also know the correlations of the predictors with the criterion variable.

Also, the authors use the terms “independent” and “dependent” variables – since this is a correlational study, they need to use predictor and criterion variable—the analysis conducted does not allow them to infer causality. Also, no collinearity diagnostics were included.  

“Multiple linear regression analyses revealed that social frailty had the strongest negative association with life satisfaction (β=-0.267, p=<.001), adjusted for sociodemographic and health-related characteristics (Table 3). However, cognitive function was significantly positively associated with life satisfaction adjusted for ??.  It is not clear what the word “adjusted” means—did they first enter sociodemographic and health-related variables, and then all of the predictor variables—if they did, they need to provide more details.

The conclusion does not say much and can be deleted.

I suggest a statistician needs to examine the analyses conducted.

Author Response

We appreciate the time and effort spent by the reviewers to provide feedback and helpful suggestions for improving the article. We carefully considered the reviewers’ comments and made numerous edits to the manuscript and highlights. We responded to each of the reviewers’ comments and outlined my manuscript revisions in the included Response to Reviewers document. We are hopeful that the revised manuscript will be worthy of publication in International Journal of Environmental Research and Public Health. We thank the editor and the reviewers for the thoughtful comments and suggestions. Our responses and a summary of changes made to the manuscript are attached.

Thank you.

Reviewer 2 Report

The paper is a cross-sectional study using Korean national survey data to understand associations of social frailty with measures of physical, cognitive, psychological variables and overall life satisfaction. It showed the prevalence of social frailty in the sample, and the various risks the population embodies including nutritional risk, depression, low cognitive function and poor life satisfaction.

Strengths:

Large population study of 10,081 lives. Builds upon previous literature and adds to the body of evidence for social frailty and pilots the association with life satisfaction. Very well done study.

Weaknesses:

The sample groups are skewed with "Robust" having a low n compared to the other groups, but this may represent the true distribution given the parameters for opting into social prefrailty and social frailty.

Specific comments:

Page 3; Line 96: Sentence is awkward. Might help to put the word "In" for the start of the sentence "The 2017 National Survey..."

Page 3; Line 89: The wording is a bit confusing for this paragraph. I'm not sure how the GDSSF-K is related to the 2017 National Survey of Older Koreans and the two scoring systems. Did the National Survey adopt the GDSSF-K in their survey, and used a cut-off of 8 as 'depressed'? Or did the authors define the cut-off?

Page 3; Line 125-6: Sentence is awkward. Would help to reword for clarity.

Page 4; Line 142: Grammar mistake. Please reword for clarity.

Page 4; Line 157: In Table 2, is there a p-value for Health Domain?

Page 6; Line 208-9: "The World Health..." sentence is awkward. Would help to reword for clarity.

Author Response

(The authors gave the same response as above.)

Round 2

Reviewer 1 Report

Review of Revised version on social frailty.

The revised version is improved but needs more work. Please see below for suggested changes.

  1. p. 2 lines 54-58. The aim of this study, conducted among community-dwelling older adults in South Korea, was to: 1) the general characteristics of older adults and the differences of social frailty in relation to these characteristics; participant’s levels of nutritional status, depression, cognitive function, and life satisfaction 2) the relationships among nutritional status, depression, cognitive function, life satisfaction and 3) clarify the health-related factors associated with a life satisfaction in older adults. (Delete a from a life satisfaction))

Comment: aim should be aims. The entire statement is broad and vague, and words are missing.  For example, in the following phrase a word is missing: in South Korea, was to (do what??—1) examine (??) the general characteristics…. (2) examine (??) the relationships among… , and (3) clarify (this is vague).

  1. p. 2. line 66 The authors state in their cover letter that made the following change, but it is included in the Data Analysis section. I think this belongs in the Participants section.

 “The 2017 National Survey of Older Korean’s sample was selected using a proportional two-stage stratified sampling method, which was first stratified and collected by 17 metropolitan cities and provinces across Korea and then again by neighborhood in the nine provinces and Sejong (but not in the metropolitan cities) [15]. The Ministry of Health and Welfare research team applied various weights in the raw data to ensure the accuracy of estimations. The weight of the raw data was adjusted by considering the weights for households and individuals [15]”

Line 71-72- Among  From the 10,299 respondents of the 2017 National Survey of Older Koreans, 10,081 were selected; 218 were excluded for missing responses.

  1. p.2, line 77: Provide the categories in the following statement: “Economic status was divided into five categories based on annual personal income.”
  2. p. 3. Line 104. Delete the phrase “By the way.” Also add: In the 2017 National Survey of Older Koreans [15], people scoring more than 8 points were classified as ‘depressed’. Did you use the same cut score—if not why mention this? (add people scoring)
  3. P. 3. Line 105. “There is evidence supporting the construct and 105 criterion-related validity of the GDSSF-K in [19”] (delete in).
  4. p. 3. Line 113. There is evidence supporting the validity of the MMSE-DS in [21] (delete in)

7.p. 3. Line 135.  “using mean and standard deviation (SD)—should be means and standard deviations (SDs)

  1. p. 4., line 139. “Before running the multiple regression analysis, we conducted a correlation analysis, (add ,) and the independent variables were tested for multicollinearity using Tolerance Value and Variance Inflation Factor (VIF).” (delete hyphen from multiple-regression; Change conduct to conducted.)
  2. 3.1. General Characteristics of the Study Population (are you studying a sample or the entire population?)
  3. p. 4, line 149-150. “The median age in the robust, social prefrailty, (delete ,) and social frailty group was 72.2 years, 73.9 years, and 75.6 years, respectively.”  Why report medians when you have reported and analyzed differences in means?
  4. p. 4. Line 152. Regarding lifestyle, the social frailty group engaged in less physical activity than the robust and social prefrailty groups (p=.000). According to Table 1, the variable examined was “frequency of people who reported “No Physical Activity” – thus the above statement is inconsistent with the analysis reported.

It also seems that the Robust group and Pre-frailty groups are more similar. Their differences at least on some variables may not be significant.

  1. Tables 1 and 2. E.S=Effect size, a=analysis of variance, b=Effect size f, c=Effect size ω. What ES measure was used for b? Also, the effect size reported is ω  for (c)—the measure is omega squared and not just omega. Also indicate the cutoff values used in evaluating ES with a reference. Also, indicate what ES measure was used for chi-squared tests.
  2. p. 5., lines 175-176. Table 3. Sshows that the predictors were statistically correlated. Depression and nutritional status had a high significant correlation (r= .455, p =.000) statistically. In addition, life satisfaction, which was the criterion variable, was statistically significantly correlated with all predictors. , as shown in the table
  3. p. 5. Lines 182-183. The variance inflation factor (VIF) of predictors and the tolerance of predictors were 1.427-2.749 and 0.364-0.701 respectively, which suggest the absence showed there was no significance in the of multicollinearity between the predictors

15.p. 6.  Discussion should follow the purposes stated—first discuss the differences in characteristics, then talk about correlations and predictions.

You start with “The purpose of this study was to investigate the correlation of social frailty with physical health, cognitive function, psychological health, and life satisfaction in community-dwelling older South Koreans.”  That was not the only purpose. Maybe just delete that sentence.

  1. p. 6, lines 198-201. “our study population was composed of community-dwelling older adults without any exclusion criteria like disabilities in activities of daily living or severe diseases and long-term care recipients [13]. Therefore, the major differences from previous studies are with regard to the items of the social frailty questionnaire and the study population. That no exclusionary was used should also be in the Participants section. Also, it would be better to use the word “study sample” and not “study population.”  It does not seem from your description that you took a random sample from a population. If you had studied the entire population, you would not need to use inferential statistical procedures.

Also, you need to include ES in discussing your results.  You just say, “It is necessary to pay attention to the interpretation of the results even based on the effect size [34], which was analyzed are reported in Tables 1 and 2.”  As authors, you need to do that and talk about whether the ES values provide any caution in taking the results seriously.

  1. As suggested before--the conclusion section can be deleted—it does not add anything to the discussion.

Author Response

We thank the reviewer for the thoughtful comments and suggestions. Our responses and a summary of changes made to the manuscript are listed below.

Reviewer 2 Report

All my comments were addressed appropriately.

Author Response

We thank the editor and the reviewers for the thoughtful comments and suggestions. 

Thank you for your consideration.

Sincerely.

Round 3

Reviewer 1 Report

Review of Version 2:

This is considerably improved. Just some minor points to take care of.

  1. These two statements are inconsistent: Abstract, line 13: “A total of 10,081 older adults met the inclusion criteria.” And later in the Participants section, line 76  it is stated “10,081 were selected without any exclusion criteria.”
  2. Measures section, Line 81: The authors stated, “Economic status was sorted in ascending order by annual personal income and divided into five categories so that one group contained 20% of the participants.” I think you want to state” “each group contained 20% of the participants” (and not one group). You might also indicate here that you were only interested in the bottom 20% of the group as this was the only group you used in analysis.
  3. Discussion section, line 242, “While the effective size can”—This should be effect size (not effective size)
  4. Discussion section, line 254.” Therefore, research is needed to identify the relationship between social frailty and life satisfaction in people of various races.” Probably better to use the phrase “ethnic groups” than races—given that the study is done in S. Korea.
  5. Reference “Rea, L. M., & Parker, R. A., Designing and conduction survey research: a comprehensive guide. 1992, San 311 Francisco: Jossey-Bass Publishers” (conducting, not conduction)

Author Response

We thank for your comments and suggestions. Our responses and a summary of changes made to the manuscript are attached.

Thank you for your consideration.  We look forward to hearing from you.

Sincerely,
